# Lipid Biomarkers as Predictors of Diastolic Dysfunction in Diabetes with Poor Glycemic Control

**DOI:** 10.3390/ijms21145079

**Published:** 2020-07-18

**Authors:** Dina Khedr, Mona Hafez, Jairo Lumpuy-Castillo, Soha Emam, Antoine Abdel-Massih, Fatma Elmougy, Rasha Elkaffas, Ignacio Mahillo-Fernández, Oscar Lorenzo, Noha Musa

**Affiliations:** 1Pediatric Diabetes, Endocrine and Metabolism Unit, Children Hospital, Cairo University, Cairo 12664, Egypt; mshereen57@yahoo.com (M.H.); noha.musa@yahoo.com (N.M.); 2Laboratory of Diabetes Research, Instituto de Investigaciones Sanitarias-Fundación Jiménez Díaz, Universidad Autónoma, 28040 Madrid, Spain; jairo.lumpuy@estudiante.uam.es (J.L.-C.); olorenzo@fjd.es (O.L.); 3Pediatric Cardiology Unit, Children’s Hospital, Cairo University, Cairo 12664, Egypt; smemam@gmail.com (S.E.); antoine.abdelmassih@kasralainy.edu.eg (A.A.-M.); 4Department of Clinical and Chemical Pathology, Cairo University, Cairo 12664, Egypt; fatma.elmougy@kasralainy.edu.eg (F.E.); Rasha.kaffas@kasralainy.edu.eg (R.E.); 5Statistics Department, Instituto de Investigaciones Sanitarias-Fundación Jiménez Díaz, Universidad Autónoma, 28040 Madrid, Spain; IMahillo@fjd.es; 6Spanish Biomedical Research Centre on Diabetes and Associated Metabolic Disorders (CIBERDEM) network, 28040 Madrid, Spain

**Keywords:** type-I diabetes, cardiac dysfunction, HDL, triglycerides, cholesterol

## Abstract

Uncontrolled type-1 diabetes (T1DM) can lead to dyslipidaemia and albuminuria, which may promote cardiovascular injuries. However, some lipidemic factors could be useful in predicting cardiac dysfunction. Seventy-eight adolescents under insulin treatment due to a 6-year history of T1DM and were retrospectively examined. Glycemia, lipidemia, and albuminuria were measured in addition to development of cardiovascular abnormalities Both girls and boys showed higher HbA1c and fasting blood glucose and 27.1% females and 33.3% males exhibited microalbuminuria though their plasma levels of total cholesterol (TC), triglycerides (TG), and low-density lipoproteins (LDL) and high-density lipoproteins (HDL lipoproteins were in the normal range. They exhibited a preserved systolic function, but 50% of females and 66.6% of males had developed diastolic failures. Interestingly, girls with diastolic dysfunction showed significantly lower concentrations of HDL and higher TC/HDL and TG/HDL ratios. In fact, low HDL levels (OR 0.93; 95% CI 0.88–0.99; *p* = 0.029) and high TC/HDL (OR 2.55; 95% CI 1.9–5.45; *p* = 0.016) and TG/HDL (OR 2.74; 95% CI 1.12–6.71; *p* = 0.028) ratios associated with the development of diastolic complications. The cut-off values for HDL, TC/HDL, and TG/HDL were 49 mg/dL, 3.0 and 1.85, respectively. HDL and TC/HDL and TG/HDL ratios may be useful for predicting diastolic dysfunction in girls with uncontrolled T1DM.

## 1. Introduction

Type 1 diabetes (T1DM) is characterized by a deficient insulin production caused by T-cell-mediated autoimmune destruction of pancreatic β-cells. T1DM is the predominant form of diabetes during childhood and adolescence though it can also be present in adulthood. The prevalence of T1DM among child and adolescent patients has increased worldwide, likely the result of dietary patterns and infection in early childhood and an increased prevalence of obesity [1]. According to a new classification from the American Diabetes Association (ADA), T1DM develops in three stages, starting with the presence of β-cell autoimmunity with normoglycemia (pre-symptomatic), followed by occurrence of dysglycemia (episodes of hypo- and hyperglycemia) and ending with insulin deficiency (symptomatic disease) [2]. The major cause of morbidity and, eventually, premature mortality in subjects with T1DM is cardiovascular disease (CVD) [3]. In population-based studies, between 14% and 45% of children with T1DM present two or more CVD risk factors including hypertension, dyslipidaemia, inflammation, obesity, and nephropathy [4]. CVD events are more common and occur 10 to 15 years earlier in patients with T1DM than in non-diabetic patients [5]. However, there is a lack of diagnosis and prognosis for CVD in T1DM subjects, and even within the first decade of diagnosis, children and adolescents with T1DM have a high occurrence of heart failure and impaired cardiac function subsequent to coronary artery disease, hypertension, and/or diabetic cardiomyopathy (DCM) [6]. Furthermore, the prevalence of CVD differs depending on the duration of T1DM, age, sex, and race/ethnicity of individuals [7]. In particular, girls with T1DM may have poor long-term clinical outcome (insulin requirements, glycemic control, and CVD) and lower generic and disease-specific quality of life than boys [8,9,10].

In addition, the ADA and European Association for the Study of Diabetes (EASD) have recently advocated for improved management of T1DM and its associated cardiovascular risk factors [11]. Paediatric echocardiography has clearly become the primary tool for describing and characterizing cardiac function in adolescents with and without heart disease. Doppler echocardiography has demonstrated the contribution of functional and structural changes in the T1DM heart. In these patients, diastolic dysfunction has been observed as an early predictor of systolic damage, and pathophysiologic manifestations includes coronary microvascular or macrovascular injury, myocardial fibrosis, and autonomic dysfunction [12]. However, cardiac contractility and active relaxation might be also deteriorated [13]. With the use of advanced echocardiographic techniques based on deformation imaging like tissue Doppler imaging (TDI), subclinical cardiopathy may be diagnosed [14]. Tissue Doppler imaging is designed to characterize low-velocity, high-amplitude signals from myocardial motion for quantification of global and regional myocardial contractile and relaxing functions [15]. Once detected, although intensive control of glycemia may not delay CVD in T1DM adolescents, chronic hyperglycemia could induce dyslipidaemia, endothelial dysfunction, arterial thickness and stiffness, autonomic neuropathy, left ventricle hypertrophy, and ventricular dysfunction [7,16]. Thus, mostly findings support the recommendation that optimal early glycemic control promotes long-term benefits against CVD [6]. However, insufficient therapy adherence and inadequate diets may lead to dysregulation in glucose homeostasis [17,18]. In this work, we analyzed the association of specific plasma and urine parameters with the development and prediction of cardiac dysfunction in a population of T1DM patients with poor glycemic control.

## 2. Results

### 2.1. Characterization of the Uncontrolled T1DM Population

We retrospectively studied a population of young T1DM patients (*n* = 78) under insulin therapy (Appendix A). After six years of treatment, we analyzed anthropometric and plasma/urine parameters in females (*n* = 48) and males (*n* = 30) (Table 1). On average, girls were 14.19 years of age, with debut of T1DM at the 6.0 years of age. Boys were 14.99 years-old, with T1DM diagnosis at 7.5 years of age. Both girls and boys presented a T1DM duration (and treatment) of 6.0 years. However, the insulin therapy they received may have not been appropriated for their level of glycemic control [19]. Uncontrolled hyperglycemia (glycated haemoglobin (HbA1c) ≥ 7.5%) was present in 85.4% of girls showed, with a mean HbA1c level of 10.12% in the last year (9.55% as mean for the previous three years) and their fasting blood glucose (FBG) was 138.0 mg/dL (Table 1). Their urinary albumin-to-creatinine ratio (A/C) was 20.0 mg/g (20.5 mg/g, as median of the last three years), and thus, 22.9% of females exhibited micro-albuminuria (A/C = 30–300 mg/g) [20] and 2.1% had macro-albuminuria (A/C > 300 mg/g) (27.1% and 0% in the last three years, respectively). According to the established guidelines [21,22], girls did not display established dyslipidaemia or overweight/obesity (Table 1). Their levels of total cholesterol (TC) were 172.0 mg/dL, low-density lipoprotein cholesterol (LDL) was 106.54 mg/dL and they had a high-density lipoprotein cholesterol (HDL) of 50.0 mg/dL (171.0, 101.0 and 49.0 mg/dL, as averages of the last three years, respectively). Concerning to triglycerides (TG), girls showed a concentration of 90.0 mg/dL in the latest year (86.0 mg/dL as median for three years), and thus the risk scores for atherosclerosis (TC/HDL ratio) and that for insulin resistance (TG/HDL ratio) were 3.46 and 1.67, respectively (3.57 and 1.84 for the last three years), which fell within the normal ranges [23,24,25]. Body mass index (BMI) was 21.25 kg/m^2^, with BMI-SDS of 0.59 and height-SDS of −0.77, and their waist circumference (WC) was 71.77 cm (Table 1).

Similarly, 86.7% of males showed poor glycemic control (HbA1c ≥ 7.5 mg/dL) [19]. Their HbA1c levels were 9.58% (8.98% in the last three years) and FBG was 127.0 mg/dL (Table 1). The A/C was 23.0 mg/g (16.0 mg/g in the last three years), and thus 33.33% and 3.3% of boys exhibited micro-albuminuria and macro-albuminuria, respectively (33.33 and 0% in the last three years) [20]. Like girls, boys did not show dyslipidaemia or overweight/obesity [21,22]. They exhibited TC levels of 163.5 mg/dL and had an LDL of 98.32 mg/dL and HDL of 43.5 mg/dL (169.5, 96.0 and 49.0 mg/dL as averages of the last three years, respectively). In boys, the median of TG levels was 83.5 mg/dL (80.0 mg/dL in the last three years) and thus the ratios for TC/HDL and TG/HDL were 3.44 and 1.71, respectively (3.39 and 1.72 in the last 3 years), falling also within the safety ranges [23,24,25]. The BMI was 20.61 kg/m^2^, with BMI-SDS of 0.51 and height-SDS of −0.80, and the WC was 71.06 cm (Table 1). Therefore, non-significant differences in (uncontrolled) glycemia, albuminuria, and lipidemia were found between boys and girls with similar age, BMI, and duration of T1DM.

### 2.2. Cardiovascular Complications Associated with T1DM

Type-1 diabetic patients exhibit increased risk for cardiovascular diseases [27]. Consequently, we studied whether T1DM patients developed hypertension and/or cardiac dysfunction. Using sphygmomanometry, we found that systolic blood pressure (SBP) was 112.16 mmHg in girls and 119.16 mmHg in boys, whereas diastolic blood pressure (DBP) was 76.08 mmHg for the former, and 76.06 mmHg for the later (Table 2 and Appendix A). These data indicate that vascular pressure was not altered [19]. However, cardiac performance could have been affected. The systolic function was evaluated by 2D-Echo/Doppler. On average, T1DM females showed a fractional shortening (FS) of 39.0%, and an ejection fraction (EF) of 73.0% (Table 2). Girls also exhibited a left ventricle end-diastolic volume index (LVEDVI) of 72.0 mL/m^2^. Males had an FS of 43.0%, EF of 77.5%, and LVEDVI of 71.0. According to previous parameters described for non-pathological systolic function in adolescents (FS > 38, EF > 68 and LVEDVI < 77 mL/m^2^ [28,29], these data suggest that systolic function could be preserved in young T1DM subjects [30]. Of interest, females showed a significant reduction in FS and EF compared to males (Table 2).

In addition, diastolic function could be impaired. Tissue Doppler imaging (TDI) enables more precise evaluation of myocardial movement dynamics and detection of discrete lesions like LV and RV diastolic dysfunction, which can be challenging to detect using a classic echocardiograph [15]. Thus, by TDI, the left ventricle E/e’ ratio, an indicator of LV diastolic function, was 8.0 in T1DM females (Table 2). In addition, the tricuspid e’/a’ ratio, which is a parameter for right ventricle (RV) diastolic function, it was 1.75. In boys, the E/e’ and e’/a’ were 10.0 and 1.55, respectively. Consequently, and following described limit values for diastolic dysfunction in adolescents (E/e’ > 6.94 [28] and/or e’/a’ < 1.28 [31]), both girls and boys with T1DM exhibited diastolic failure. In particular, LV diastolic abnormalities were present in 41.7% of girls and in 63.3% of boys, whereas RV diastolic dysfunction was observed in 14.6% of girls and in 13.3% of boys.

### 2.3. Prediction of Ventricular Diastolic Dysfunction in T1DM Patients

Next, we further studied whether some plasma or urine biomarkers could associate with the development of cardiac damage (i.e., diastolic dysfunction) in this population. Female and male patients were classified according to the presence of ventricular (LV and/or RV) diastolic injury assessed by TDI. Fifty percent population of females (*n* = 24) and 66.6% of males (*n* = 20) exhibited diastolic dysfunction (Table 3). Interestingly, the levels for the previous year and the 3-year average Hb1Ac, and FBG did not associate with the development of diastolic alteration either in girls or boys (Table 3). Similar results were obtained for micro- and macro-albuminuria, and also for plasma levels of TC, LDL, and TG, and BMI and WC. As expected, neither blood pressure nor parameters of systolic function associated with diastolic dysfunction (Table 1).

However, there was a significant positive association between HDL levels and the TC/HDL and TG/HDL ratios with diastolic damage only in T1DM girls (Table 3). These girls exhibited a 3-year average HDL concentration of 46.0 mg/dL, while those without diastolic failure showed a level of 54.0 mg/dL (*p* < 0.02). In addition, in females, the 3-year average for TC/HDL and TG/HDL ratios were significantly higher in those with diastolic dysfunction (3.93 and 2.09, respectively) than in girls without diastolic failure (3.22 and 1.58, respectively) (*p* = 0.02 and *p* = 0.01, respectively). Similar results were seen for quantifications of these parameters in the last year (Table 3). These data suggest that HDL levels and their ratios with TC and TG may predict diastolic dysfunction in girls with uncontrolled T1DM. In this sense, we tested potential correlations (by Pearson’s) among the diastolic function parameters (LV-E/e’ and RV-e’/a’) and lipid levels (HDL and the TC/HDL and TG/HDL ratios) in girls. Interestingly, the LV-E/e’ ratio showed a significant positive correlation with TC/HDL and a negative correlation with HDL (Figure 1A). However, correlation coefficients were weak for both cases (*r*^2^ = 0.11−0.18) (Figure 1B).

Therefore, we studied their potential associations by applying a model of binary logistic regression and ROC curves. In fact, the univariate analysis confirmed that low levels of 3-year average HDL were associated with the development of diastolic complications in T1DM girls (OR 0.93; 95% CI 0.88–0.99; *p* = 0.029) (Figure 2A). In addition, the elevated ratios of TC/HDL and of TG/HDL associated with diastolic failure in these patients (OR 2.55, 95% CI 1.9–5.45, *p* = 0.016 and OR 2.74, 95% CI 1.12-6.71, *p* = 0.028, respectively). In addition, by ROC curve analysis, the cut-off value for the 3-year average of HDL was 49.0 mg/dL (with 71% sensitivity and 71% specificity), of TC/HDL was 3.46 (71% sensitivity and 62% specificity), and that, for TG/HDL, it was 1.85 (67% sensitivity and 83% specificity). Similar data were found for the last yearly measurement (HDL; 51 mg/dL (with 71% sensitivity and 67% specificity), TC/HDL; 3.5 (71% sensitivity and 75% specificity), and TG/HDL; 1.67 (71% sensitivity and 71% specificity)) (Figure 2B).

## 3. Discussion

T1DM adolescents with six years of diabetes duration and insulin treatment showed poor glycemic control and alterations in some plasma, urine, and cardiovascular parameters. In particular, both girls and boys with T1DM exhibited a preserved ventricular systolic dysfunction, but presented microalbuminuria and LV and RV diastolic failure. Interestingly, although their lipid profiles were within the safety limit, changes in plasma HDL levels and in the TC/HDL and TG/HDL ratios predicted diastolic dysfunction only in females.

Diabetes management for children and adolescents cannot be extrapolated from adult diabetes care. Once diabetes is detected, the ADA recommends that most adolescents with T1DM should initially follow a regimen of MDI of basal/bolus insulin [19]. However, a lack of treatment adherence, monitoring, and follow-up of anti-diabetic therapy and/or unsuitable diets can often provoke suboptimal glycemic control and adverse consequences [18,32]. In fact, our patients did not reach safe levels on plasma HbA1c and FBG, and showed renal glomerular injury in the form of proteinuria. Chronic hyperglycemia can stimulate the release of reactive oxygen species, inflammatory cytokines, and growth factors, from monocytes and vascular cells, which damage the endothelial function and favor premature atherogenesis [33]. Thus, a worse glycemic control has been associated with inflammation and atherogenesis in T1DM subjects [34]. Moreover, the increased CD40 ligand expression and platelet-monocyte aggregation contribute to the accelerated rate of atherothrombosis in these patients [35]. In the kidney, this endothelial injury can alter the glomerular filtration barrier, and initiate the diabetic microalbuminuria [36]. In our study, 27.1% and 33.3% of girls and boys, respectively, presented microalbuminuria. According to the literature, the prevalence of microalbuminuria in adolescents with T1DM varies from 10% to 40%, but only 5% to 10% of them manifest persistent kidney damage due to improvements in renal haemodynamics associated with pubertal growth and development [37]. However, T1DM patients with albuminuria are at greatly increased risk of having subclinical abnormal cardiovascular function compared to patients without albuminuria [38] (Figure 3).

In our case, after six years of uncontrolled T1DM and microalbuminuria, girls and boys were normotensive and exhibited preserved systolic function, but they showed an evident diastolic dysfunction (by 2D-Echo/Doppler and TDI). The existence of DCM has been proposed as evidence of myocardial dysfunction in diabetic patients in the absence of ischemic, valvular or hypertensive heart disease [39]. Despite scarce evidence of established echocardiographic ranges for cardiac dysfunction in adolescence [40], more than 40% and 60% of girls and boys, respectively, showed LV diastolic abnormalities, whereas RV diastolic injury was observed in 14.6% of girls and in 13.3% of boys. Impairment of diastolic but not systolic function was also previously noticed in young with T1DM [41]. DCM is defined as ventricular dysfunction initiated by alterations in cardiac energy substrates in the absence of coronary artery disease and hypertension [39]. Functionally, DCM is characterized by diastolic dysfunction, which manifests as a defect in left ventricular relaxation leading to increased pressures and a subsequent impaired filling during diastole. Interestingly, a decrease of testosterone and estrogen levels could play a key role in the progression of DCM and CVD [42]. Perhaps, systolic function could be detected in more advanced stages of the disease as a consequence, at least, of diastolic anomalies. Indeed, other authors described both diastolic and systolic failures in middle-aged T1DM patients [15]. Thus, persistent hyperglycemia can damage heart function indirectly by vascular injury but also directly on the myocardium. Indeed, the excessive glucose can induce hypertrophy and fibrosis in cardiac cells similar to those in the vascular niche [43].

Interestingly, however, we found non-significant differences in (uncontrolled) glycemia, albuminuria, and lipidemia between boys and girls, our females revealed significantly lower FS and EF than males. The relative impact of CVD on T1DM compared with the general population may be much higher for women than for men. Women with T1DM had a 40% greater excess risk of all-cause mortality, and twice the excess risk of fatal and non-fatal vascular events, compared with men with T1DM. In a recent meta-analysis, the T1DM-associated mortality ratio attributable to CVD was 5.7 for men and 11.3 for women [44]. In addition, a very large German study of 33.333 children and adolescents with T1DM found that girls displayed higher prevalence of hyperglycemia and dyslipidaemia than boys [45]. Likely, sex differences in youth with T1DM could be influenced by pubertal stages and subsequent metabolic and hormonal control, and by fat distribution patterns associated with insulin resistance or deregulated lipids (i.e., HDL) [9,46]. A complex relationship linking CVD, fat, and diabetes has been evidenced. The prevalence of hypertension and other CVD increases abruptly with increasing body weight, which plays an essential role in the development of insulin resistance and diabetes [47]. The underlying molecular mechanisms are still uncovered. In this regard, uncontrolled glycemia may also lead to dyslipidaemia. In the SEARCH for Diabetes in Youth study, adolescents with poor glycemic control exhibited alteration in their plasma lipid, but an improvement of glucose control over a two-year period was associated with a more favorable lipid profile [48] (Figure 2). The HDL fraction is of particular current interest because its metabolism can be modified in T1DM due to abnormal lipoprotein lipase and hepatic lipase activities related to exogenously administered insulin [49]. In addition, the increased levels of plasma haptoglobin in T1DM can promote the interaction of macrophages with HDL, reducing HDL availability [50]. Thus, in the Diabetes Control and Complications Trial (DCTT), patients with poorly controlled insulin-treated T1DM were found to have low HDL levels [51]. Consequently, a TC/HDL higher than 3.8 has been suggested to predict cardiovascular injuries (i.e., ischemic heart disease) in adolescents [23]. Similarly, a TG/HDL ratio greater than 3 has been considered a marker of insulin resistance in adults, and that a ratio over 2.05 may be a good indicator of insulin resistance in youth [24,52]. Interestingly, average HDL concentration over the last three years was significantly lower (46.0 mg/dL) only in girls with diastolic dysfunction when compared to females without diastolic failure. Moreover, their levels of TC/HDL and TG/HDL were higher (3.93 and 2.09, which fall into risk ranges of atherosclerosis and insulin resistance, respectively) than in those girls without diastolic damage. Lower levels of HDL have been also associated with coronary artery disease in women [53]. Therefore, 3-year quantifications of HDL and both TC/HDL and TG/HDL significantly associated with a later diastolic failure in T1DM girls, and thus these parameters could be used to predict of diastolic dysfunction (Figure 2). In particular, levels lower than 49.0 mg/dL for HDL, and higher than 3.46 and 1.85 for TC/HDL and TG/HDL ratios, respectively, may be considered as risk factors of diastolic damage in these subjects. Importantly, low levels of HDL have been consistently associated with increased risk of diabetes. HDL may contribute to the pathophysiology of T1DM through direct effects on plasma glucose levels. In particular, HDL stimulates pancreatic β-cell insulin secretion and modulates glucose uptake in skeletal muscle [54]. However, not only changes in the amount, but also in function of HDL are frequent in these patients. In this sense, genetically reduced HDL did not associate with increased risk of diabetes [55]. HDL may become dysfunctional, losing its anti-atherogenic, anti-oxidative, and anti-inflammatory properties, because of alterations in its relative composition of lipids and proteins or in related enzymes (i.e., paraoxonase-1, lipoprotein-associated phospholipase-11) [56]. Indeed, some HDL particles may not be efficient for reverse cholesterol transport due to the poor ability to release its apolipoprotein A-I [57]. In this sense, HDL type-2 (but not HDL type-3C) were inversely associated with insulin resistance and type-II diabetes [58]. Thus, analysing both HDL function as well as HDL types and levels may offer a better assessment of CVD risk [59]. In fact, HDL function decreased early after the onset of T1D and persisted over 5 years in children with T1D, contributing to an enhanced CVD risk [57]. Nevertheless, the correlation between types and function of HDL with cardiac dysfunction needs to be demonstrated in both males and females with T1DM, and then, it could be proposed also as prognosis biomarker.

Finally, in adolescents with uncontrolled T1DM, continuous subcutaneous insulin infusion could improve glycemia [19], but only glycemic control may be unlikely to normalize dyslipidaemia and subsequent potential cardiac dysfunction. In this sense, the guidelines for Cardiovascular Health and Risk Reduction in Children and Adolescents [26] recommends optimizing glucose control and administrating nutrition therapy that restricts saturated fat to 7% of total calories and dietary cholesterol to 200 mg/day, which is safe and does not interfere with normal adolescent growth and development of the adolescents. The statin administration could be also considered in non-responding youth (LDL >130 mg/dL with at least one CVD risk factor).

## 4. Material and Methods

### 4.1. Study Population

This retrospective cohort study included 78 adolescents (12–18 years of age) with a T1DM diagnosis following the criteria of the Paediatric Diabetes, Endocrine and Metabolism Unit, at the Cairo University Children’s Hospital (Hb1Ac > 7.5% and fasting glucose > 126 mg/dL). Patients with type-2 diabetes or monogenic diabetes, congenital heart diseases or arrhythmias, acute or chronic systemic disease potentially affecting cardiac function (i.e., hypertension, renal failure), and patients under medications known to affect cardiac physiology (i.e., beta blockers, angiotensin converting enzyme inhibitors, diuretics, anti-arrhythmic) were excluded from the study. Both girls and boys were treated with a multiple daily injection (MDI) regimen of insulin (basal or prandial) for six years (Appendix A), with a mean requirement of 0.7 ± 0.3 unit/kg/day. Patients were recruited in June 2015 and followed up until June 2017, when a cardiologist assessed the cardiac function. The study protocol was approved by the Research Ethics Committee of Cairo University Hospitals (Ref. I-200317-06/2017) and the informed consents was obtained from the participants’ legal guardians before enrolment. This work was carried out in accordance with The Code of Ethics of the World Medical Association (Declaration of Helsinki).

### 4.2. Anthropometric, Plasma, and Urine Parameters

A full clinical history was obtained for each patient including data on age, sex, onset of diabetes, diabetes duration, daily insulin requirements and blood and urine readings. Physical examination by an endocrinologist included a blood pressure (BP) assessment (using mercury sphygmomanometer at 3 different times within 2 weeks), which was plotted on Egyptian BP curves [60]. Anthropometric data (height and height-standard deviation score (SDS), body mass index (BMI, as body weight (kg)/height (m^2^)), BMI-SDS and waist circumference (WC)) were also plotted on Egyptian curves of pubertal staging [60,61]. Blood samples were collected from T1DM girls and boys at three different times (in June 2015, 2016 and 2017) of the study (Appendix A). Blood-EDTA samples were immediately centrifugated (20 min, 2500 rpm, 4 °C), and plasma was stored at −80 °C until use. Fasting blood glucose, HbA1c (Tosoh G8, Tosoh Bioscience, Japan), TC, TG, LDL and HDL (Beckman coulter, Beckman, Germany) were evaluated in the Clinical Pathology Department at the Cairo University Children’s Hospital. Urine samples were also collected in June 2015, 2016 and 2017 to establish the albumin to creatinine ratio (A/C) (BN Prospec, Siemens, Germany). Presence of micro-albuminuria or macro-albuminuria was determined in patients with an A/C of 30–299 mg/g or ≥300 mg/g, respectively.

### 4.3. Echocardiographic Studies

Echocardiography was performed in girls and boys at the end of the study by using conventional trans-thoracic two dimensional (2D) Echo/Doppler (M-mode) and Tissue Doppler Imaging (TDI) with the General Electric Vivid 7 Ultrasound System (Vingmed model N-3190, Horten, Norway) according to the guidelines of the American Society of Echocardiography [62]. By means of 2D-Echo/Doppler, M-mode measurements were made at the tips of the mitral valve leaflets in the parasternal long axis view. Left ventricular (LV) dimensions were examined in systole and diastole, and then, fractional shortening (FS) and ejection fraction (EF) were estimated [63]. In addition, LV end-diastolic volume index (LVEDVI) was calculated by dividing left ventricular end diastolic volume by body surface area. Using TDI, the early filling (E) and early diastolic mitral annular velocity (e’) ratio (E/e’) was assessed at the mid-oesophageal four-chambers view with the pulsed-wave Doppler in basal segments of the LV lateral wall and septal wall for early detection of diastolic LV dysfunction [28]. For RV diastolic function, we estimated the early (e’) and late diastolic tricuspid annular velocities (a’) ratio (e’/a’), as previously documented [64].

### 4.4. Statistical Analysis

Qualitative variables were included as absolute and relative frequencies. Associations between qualitative variables were studied by the chi-square test, or by the Fisher’s exact test in those cases in which the chi-square distribution was not appropriated. Quantitative variables were summarized as mean values and standard deviation, or by median and interquartile range, depending on the symmetry of the data distribution. Normality of quantitative variables was analyzed by the Kolmogorov–Smirnov test. Variables with normal distribution were compared using a Student’s *t*-test, while variables with non-normal distribution were compared using the Mann–Whitney U test. Associations between quantitative variables were studied by the Pearson’s correlation coefficient. In order to identify potential predictors for ventricular diastolic dysfunction, logistic regression models and receiver operating characteristic (ROC) curves were used. Logistic models were summarized by the odds ratio, its 95% confidence interval, and *p*-value. ROC curves showed the area under the curve with the 95% confidence interval and the cut-off point obtained by Youden’s criterion, with respective sensitivity and specificity values. Statistical analyses were performed using the statistical package for social science (SPSS), version 25.0 (IBM, New York, NY, USA).

## 5. Conclusions

Uncontrolled hyperglycemia could promote albuminuria, dyslipidaemia, and cardiac diastolic dysfunction in adolescents with DM1. Interestingly, a less favorable lipid profile could explain the lack of cardiovascular protection in females with T1DM. In this sense, alterations in plasma HDL and its related factors, TC/HDL and TG/HDL, could be useful in predicting diastolic dysfunction in T1DM girls. A value information could be also provided by the evaluation of HDL function and types in these patients. Nevertheless, new therapeutic strategies are needed to improve the evolution of both glycemia and lipidemia in subjects with T1DM to avoid or delay a significant CVD.

## 6. Limitations of the Study

Undoubtedly, the number of patients with DM1 (*n* = 78; 48 girls and 30 boys) was limited and, therefore, the results obtained should be confirmed in a larger population. A larger study should also include a more extensive 2D-Echo/Doppler and TDI analysis, evaluating the effect of T1DM and its lack of glycemic control on other key parameters for ventricular function and size (e.g., ventricular diameters, volumes, and wall-thicknesses in systole and diastole, ventricular mass, ejection volumes and cardiac output, regional systolic and diastolic function, aortic and trans-mitral flows) and for the cardiac hemodynamics. In this sense, the speckle tracking echocardiography (STE) can objectively quantify with more sensitivity the general and regional myocardial function independently of the angle of myocardial insonation [65]. STE also obtains myocardial deformation data in three dimensions (radial, circumferential, and longitudinal) by automatic measurement of the distance between two points of each segment of the LV during the cardiac cycle, and it could be used to evaluate rotational mechanics of the LV.

## Figures and Tables

**Figure 1 ijms-21-05079-f001:**
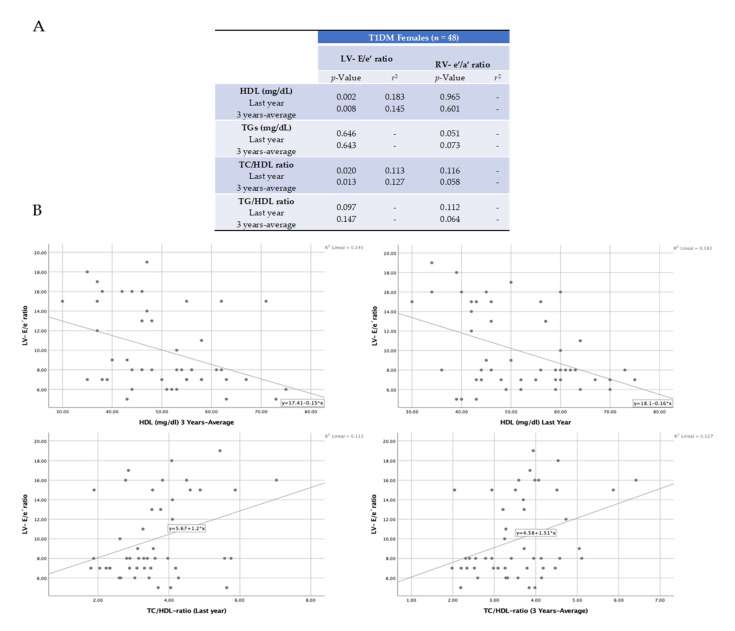
Pearson’s correlations for diastolic dysfunction and lipid biomarkers in girls. The Pearson’s correlation coefficient was used to measure the strength of the relationship between the ratios of LV-E/e’ and RV-e’/a’, and plasma levels of HDL, TC/HDL, and TG/HDL (**A**). Significant positive and negative correlations were found for TC/HDL and HDL, respectively, with LV-E/e’ ratio. The resultant r squared is also indicated (**B**).

**Figure 2 ijms-21-05079-f002:**
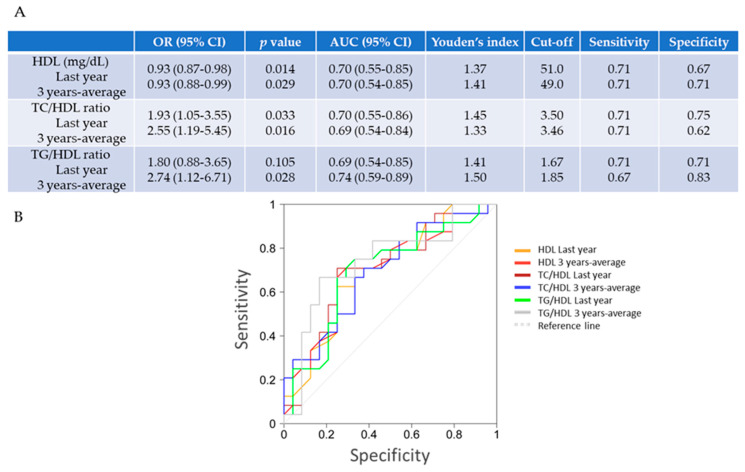
Independent predictive factors of diastolic dysfunction in T1DM females. (**A**,**B**) By stepwise regression analysis, low levels of HDL and a high of TC/HDL could predict the presence of ventricular diastolic dysfunction (by TDI) in uncontrolled T1DM girls. Using a Receiver Operating Characteristic (ROC) curve (bottom) and applying Youden’s index, the (last-year) plasma levels of HDL lower than 51 mg/dL, and TC/HDL and TG/HDL higher than 3.5 and 1.67, respectively, could be used as threshold values for prediction of diastolic failure. For a 3-year measurement, levels of HDL under 49.0 mg/dL and of TC/HDL and TG/HDL over 3.46 and 1.85, respectively, might predict diastolic dysfunction in T1DM girls. OR, odds ratio; AUC, area under the curve; CI, confidence interval.

**Figure 3 ijms-21-05079-f003:**
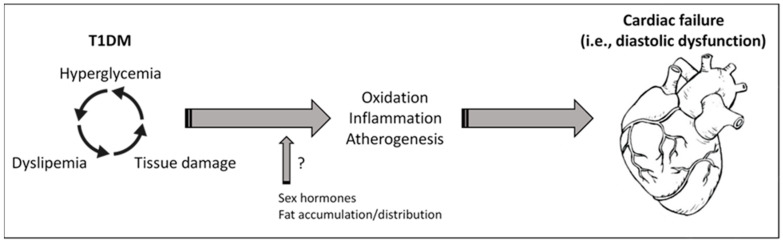
Hyperglycemia, dyslipemia, and cardiac dysfunction in T1DM. Uncontrolled glycemia in patients with T1DM may promote tissue injuries (i.e., in vessel, kidney) and alterations in plasma lipid, such as TC, TG, and HDL. In turn, dyslipemia can enforce tissue damage and reduce its glucose assimilation. In consequence, increased oxidation, inflammation, and atherogenesis could lead to cardiovascular abnormalities and heart dysfunction. Sexual hormones and the body fat accumulation and distribution may play key roles in these responses, affecting differentially to females and males. Of interest, HDL and the ratios of TC/HDL and TG/HDL may serve as prognostic biomarkers for cardiac failure.

**Table 1 ijms-21-05079-t001:** Characterization of the T1DM population.

	T1DM Males (*n* = 30)	T1DM Females (*n* = 48)	*p* Value	Reference Ranges		
				Healthy	Borderline	Abnormal
Age (years)	14.99 ± 2.59	14.19 ± 2.29	0.15	*-*	*-*	*-*
T1DM onset (age)	7.5 (4.13)	6.0 (3.88)	0.35	*-*	*-*	*-*
Duration of T1DM (years)	6.0 (3.13)	6.0 (3.75)	0.73	*-*	*-*	*-*
HbA1c (%) Last year 3 years-average	9.58 ± 2.14 8.98 ± 1.48	10.12 ± 2.49 9.55 ± 1.74	0.33 0.13	<7.5	7.5–8.0	≥8.0 [19]
FBG (mg/dL) Last year	127.0 (64.25)	138.0 (39.75)	0.70	<100	≥100–126	≥126 [26]
A/C ratio (mg/g) Last year 3 years-average	23.0 (55.75) 16.0 (27.0)	20.0 (22.25) 20.5 (15.98)	0.13 0.28	<30	30–299	≥300 [20]
Micro-albuminuria (%) Last year 3 years-average	33.33 33.33	22.90 27.10	0.31 0.55	-	-	-
Macro-albuminuria (%) Last year 3 years-average	3.3 0	2.1 0	0.73 -	-	-	-
TC (mg/dL) Last year 3 years-average	163.5 (64.25) 169.5 (50.25)	172.0 (36.5) 171.0 (39.25)	0.16 0.24	<170	170–199	≥200 [21]
LDL (mg/dL) Last year 3 years-average	98.32 ± 26.73 96.0 (39.75)	106.54 ± 26.77 101.0 (24.5)	0.19 0.07	<110	110–129	≥130 [21]
HDL (mg/dL) Last year 3 years-average	43.5 (18.75) 49.0 (11.25)	50.0 (17.0) 49.0 (14.5)	0.21 0.61	>45	40–45	≤40 [21]
TGs (mg/dL) Last year 3 years-average	83.5 (37.0) 80.0 (28.25)	90.0 (33.5) 86.0 (29.75)	0.81 0.35	<90	90–129	≥130 [21]
TC/HDL ratio Last year 3 years-average	3.44 (1.49) 3.39 ± 0.86	3.46 (1.28) 3.57 ± 0.96	0.70 0.40	<3.8	-	>3.8 [23]
TG/HDL ratio Last year 3 years-average	1.71 (1.08) 1.72 ± 0.63	1.67 (1.0) 1.84 ± 0.76	0.86 0.47	<2.05	-	>2.05 [24]
BMI (kg/m^2^)	20.61 ± 2.64	21.25 ± 3.69	0.40	-	-	≥85th perc. [22]
BMI-SDS	0.51 ± 0.95	0.59 ± 1.06	0.73	-	-	-
Height-SDS	−0.80 ± 1.25	−0.77 ± 1.16	0.89	-	-	-
WC (cm)	71.06 ± 6.86	71.77 ± 8.93	0.71	-	-	≥90th perc. [22]

Anthropometric (age, T1DM onset and duration, BMI, BMI-SDS and Height-SDS), plasmatic glycated haemoglobin (HbA1c), fasting blood glucose (FBG) and the lipid profile (total cholesterol, TC; LDL and HDL lipoproteins, and triglycerides, TG) and urinary (the albumin/creatinine ratio (A/C) and micro-/macro-albuminuria prevalence) parameters, were evaluated in males (*n* = 30) and females (*n* = 48) with T1DM. Some factors were also estimated as average values for the previous three years. The *p*-value indicates differences between both groups. The reference ranges for adolescents (healthy, borderline, and abnormal) for some parameters are also indicated.

**Table 2 ijms-21-05079-t002:** Blood pressure and LV and RV function in uncontrolled T1DM subjects.

	T1DM Males (*n* = 30)	T1DM Females (*n* = 48)	*p* Value	Reference Ranges		
				Healthy	Borderline	Abnormal
SBP (mm Hg)	119.16 ± 8.59	112.16 ± 9.32	0.001	<120	120–129	≥130 [19]
DBP (mm Hg)	76.06 ± 7.66	76.08 ± 8.86	0.99	<80	-	≥80 [19]
2D-Echo/Doppler						
FS (%)	43.0 (7.0)	39.0 (8.0)	0.03	>38	-	<38 [28]
EF (%)	77.5 (12.25)	73.0 (14.75)	0.03	>68	-	<68 [28]
LVEDVI (mL/m^2^)	71.0 (6.25)	72.0 (4.75)	0.58	<77	-	>77 [29]
TDI						
LV—E/e’ ratio	10.0 (6.25)	8.0 (7.75)	0.50	<6.94	-	>6.94 [28]
RV—e’/a’ ratio	1.55 (0.90)	1.75 (0.90)	0.91	>1.28	-	<1.28 [31]

Systolic and diastolic blood pressures (SBP and DBP) were evaluated in males and females with T1DM at the end of the study. In addition, systolic and diastolic ventricular function were estimated by 2D-Echo/Doppler and TDI, respectively. The reference ranges for adolescents (healthy, borderline and abnormal) for some parameters are also indicated. FS, fractional shortening; EF, ejection fraction; LVEDVI, LV end-diastolic volume index; LV-E/e’, left ventricle early filling (E) and early diastolic mitral annular velocity (e’) ratio; and RV-e’/a’, early tricuspid annular velocity (e’) and late diastolic tricuspid annular velocity (a’) ratio.

**Table 3 ijms-21-05079-t003:** Associations between anthropometric and plasmatic/urinary parameters with cardiac diastolic dysfunction in uncontrolled T1DM.

	T1DM Males (*n* = 30)			T1DM Females (*n* = 48)		
	Non-Diastolic Dysfunction (*n* = 10)	Diastolic Dysfunction (*n* = 20)	*p* Value	Non-Diastolic Dysfunction (*n* = 24)	Diastolic Dysfunction (*n* = 24)	*p* Value
Age (years)	16.09 ± 2.70	14.45 ± 2.42	0.10	14.75 ± 2.45	13.63 ± 2.01	0.09
T1DM onset (age)	9.0 (2.75)	5.5(5.38)	0.10	5.75 (4.38)	6.0 (4.0)	0.65
Duration of T1DM (years)	6.50 (2.50)	6.00 (3.38)	0.61	6.25 (5.50)	6.00 (3.00)	0.66
HbA1c (%) Last year 3 years-average	9.82 ± 2.61 9.2 ± 1.82	9.46 ± 1.94 8.87 ± 1.33	0.67 0.58	9.53 ± 2.72 9.24 ± 1.80	10.71 ± 2.14 9.87 ± 1.676	0.10 0.21
FBG (mg/dL) Last year	153.5 (84.5)	124.5 (51.5)	0.24	134.0 (34.0)	139.5 (63.5)	0.59
A/C ratio (mg/g) Last year 3 years-average	21.0 (77.0) 15.0 (13.50)	26.5 (45.85) 18.0 (40.5)	0.55 0.44	18.0 (16.50) 17.5 (11.25)	21.5 (37.25) 24.5 (18.75)	0.24 0.06
Micro-albuminuria (%) Last year 3 years-average	30 10	70 90	0.72 0.05	27.3 30.8	72.7 69.2	0.86 0.10
Macro-albuminuria (%) Last year 3 years-average	0 0	100 0	0.72 -	0 0	100 0	0.31 -
TC (mg/dL) Last year 3 years-average	146.0 (78.5) 139.5(69.25)	169.0 (50.75) 171.5 (41.0)	0.24 0.23	169.0 (50.0) 161.5 (48.0)	172.0 (45.75) 173 (29.5)	0.29 0.18
LDL (mg/dL) Last year 3 years-average	87.5 ± 21.71 79.0 (35.75)	103.74 ± 27.84 100.0 (53.25)	0.11 0.07	102.45 ± 28.75 100 (39.75)	110.62 ± 24.56 104.5 (19.0)	0.29 0.17
HDL (mg/dL) Last year 3 years-average	43.0 (16.0)48.0 (16.50)	45.0 (24.25)50.0 (9.5)	0.470.32	57.5 (18.5)54.0 (15.25)	45.5 (14.75)46.0 (15.25)	0.020.02
TGs (mg/dL) Last year 3 years-average	83.0 (40.0) 73.50 (21.50)	85.0 (38.25) 84.5 (38.5)	0.81 0.28	84.0 (41.5) 79.0 (30.0)	90.0 (31.25) 87.0 (29.5)	0.26 0.06
TC/HDL ratio Last year 3 years-average	3.22 (2.01) 3.46 ±1.18	3.44 (1.13) 3.36 ± 0.69	0.84 0.78	3.12 (1.24) 3.22 ± 0.78	3.79 (1.42) 3.93 ± 1.01	0.01 0.01
TG/HDL ratio Last year 3 years-average	1.74 (1.62) 1.73 ± 0.64	1.71 (0.83) 1.71 ± 0.64	0.81 0.93	1.47 (0.95) 1.58 ± 0.71	2.03 (0.91) 2.09 ± 0.74	0.02 0.02
BMI (Kg/m2)	20.79 ± 1.98	20.52 ± 2.96	0.79	21.58 ± 3.32	20.92 ± 4.08	0.53
BMI-SDS	0.40 ± 0.74	0.57 ± 1.05	0.65	0.60 ± 1.01	0.58 ± 1.14	0.95
Height-SDS	−1.13± 1.53	−0.64 ± 1.09	0.32	−0.64 ± 1.09	−0.90 ± 1.23	0.44
WC (cm)	71.1±3.6	71.05 ± 8.11	0.98	72.41 ± 7.90	71.12 ± 9.99	0.62

Glycemia, albuminuria and TC, LDL, BMI and WC did not associate with ventricular diastolic failure either in boys or girls with T1DM. However, plasma HDL levels and the TC/HDL and TG/HDL were significantly associated with development of diastolic dysfunction in girls.

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
