# Peer review of "Lipid Biomarkers as Predictors of Diastolic Dysfunction in Diabetes with Poor Glycemic Control"

_ijms, 2020, doi:10.3390/ijms21145079_

Round 1

Reviewer 1 Report

REVIEWER’S COMMENTS

for:

Manuscript entitled “Lipidemic biomarkers as predictors of diastolic dysfunction in diabetes with poor glycemic control”

Authors: Khedr D. et al.

Journal: IJMS; Manuscript ID: ijms-848379

Summary: In this current article Khedr and colleagues describe the clinical value of measuring HDL and triglyceride levels in young diabetic patients, as these markers may be used as predictors of diastolic cardiac dysfunction, in female patients in particular. The general topic is interesting, and this current manuscript aims to provide evidence-based rationale for the assessment of lipid biomarkers in early T1DM, as authors suggest it may become valuable parameters in estimating cardiovascular risk in young patients.

General comments

Overall, this reviewer is satisfied with the composition of the manuscript, as it describes applicable clinical evidences with possible benefits for physicians. Although it should be noted that authors need to clarify the novelty of this report in the view of recent literature data describing similar findings. Moreover, illustrative figures - showing the correlation between the measured parameters, and as well the echocardiographic method - are missing. In addition, a few issues addressed below would further strengthen the conclusions and improve the manuscript. Please, read my major and minor comments below.

Major comments

  1. As a fundamental method in this manuscript is echocardiography, authors need to clarify the following: In Methods 2.3, they write: “For RV diastolic function, the early filling (E) and late diastolic mitral annular velocity (A') ratio (E/A’) of the basal segments of the RV free wall was estimated.” First, there is no reference for this method, please indicate it. Second: How mitral velocities could predict right ventricular dysfunction? Is this only a misspelling (tricuspid?)? To better estimate RV function, authors should have measured tricuspid E/A, and tricuspid TVI e’ and a’ velocities. In addition, in results Table 2, authors refer Choi S-H, Eun LY, Kim NK, Jung JW, Choi JY. Myocardial Tissue Doppler Velocity in Child Growth. J Cardiovasc Ultrasound. 2016;24:40–7 (reference 23); and write that E/A’ ratio above 2.26 indicates RV diastolic dysfunction. However, this reviewer could not find this fact in the referred paper. Moreover, In the Table 2 legend, authors write “tricuspid E'/A' ratio” “as parameter for right ventricle (RV) diastolic function” - which is correct - but in the Table, there is E/A’ ratio again. Since in echocardiography, Doppler blood flow velocities are classically indicated by letters E and A, while TDI tissue velocities are indicated by commas (e’, a’ or E’, A’), the results presented here related to RV function are a lot confusing. Please, carefully revise the text, indicate precisely whether it is E/A’ or e’/a’, mitral or tricuspid, etc., and clarify this whole issue.

  1. It is known that a deficit in HDL function is a significant contributor to the enhanced CVD risk in young T1DM patients, moreover, measuring of high-density lipoprotein (HDL) function provide a better risk estimate for future CVD events than only serum levels of HDL cholesterol, see Heier et al, 2017 (doi: 10.1186/s12933-017-0570-2.). Regarding these previous findings, please, discuss the novelty and value of this current report in a short paragraph, and compare your results with literature data.

  1. Authors of this current report clearly prefer tables over figures, graphs and curves. This reviewer thinks that since the main message of this paper is a correlation between diastolic function and lipid parameters, and additional graph would be useful to strengthen data. Did authors perform further statistical analyses (correlation, r square) regarding echo values, diastolic dysfunction and lipid parameters? If yes, please insert a correlation graph to the text.

Minor comments

  1. Illustrative figures are missing from this current paper. This reviewer thinks that using figures could facilitate the understanding, e.g.: echocardiographic images.

  1. Although this review paper aims to describe clinical aspects, this reviewer thinks that inserting a short paragraph describing the pathophysiology of diabetic cardiomyopathy and diastolic dysfunction grading would be useful. Please refer: doi: 10.3389/fphys.2018.00453 and doi: 10.1186/1471-2369-14-76 , or similar articles.

  1. Although there were very few errors in grammar and punctuation, some of the sentences were phrased a bit awkwardly. Minor grammatical changes made by a professional would be valued.

In summary, this manuscript is well-written and has a major impact for the readers of this Journal. This reviewer suggests that it is acceptable for publication in IJMS, after major revisions.

Reviewer 2 Report

The role and function of HDL in patients with diabetes mellitus and the related cardiovascular risk has been recently discussed (doi: 10.1186/s12944-017-0594-3; doi: 10.2337/db14-1603; 10.3390/jcm8020253).

ROC Curves: please calculate the Youden’s index (quote J. Clin. Med. 2019;8:2061; PMID:31771147) and report the highest J value for each variable.

The complex relationship linking diabetes, fat, and cardiovascular disease has been recently examined by Shu and colleagues (doi: 10.1111/jch.13453) and should be mentioned in the discussion.

Round 2

Reviewer 1 Report

2nd round REVIEWER’S COMMENTS

for:

Manuscript entitled “Lipidemic biomarkers as predictors of diastolic dysfunction in diabetes with poor glycemic control”

Authors: Khedr D. et al.

Journal: IJMS; Manuscript ID: ijms-848379

Summary: Authors of this present manuscript clarified major issues regarding echocardiography, statistics and data presentation.

Right ventricle echo parameters were corrected, references were added and replaced. This major issue (echocardiography) has been solved.

Tables were fixed and new references were implied into the text, thus, the value of this paper significantly improved.

Authors showed correlation analyses and replaced the word “correlate” to “associate” in context of diastolic dysfunction and lipid biomarkers, which is more correct.

Minor comments

  1. Illustrative figures are still missing. I understand that quality of echo images may be poor, nevertheless, other types of figures would be possible: e.g. graphical abstract, study design, correlation graphs. These would ease understanding and I still recommend to implicate an illustrative figure into the main text.

  1. Although an English expert revised the manuscript, I still have concerns with the title. Why have authors chosen the word “lipidemic” biomarkers instead “lipid”?

Reviewer 2 Report

-
